# Nutritional Changes and Early Warning of Moldy Rice under Different Relative Humidity and Storage Temperature

**DOI:** 10.3390/foods11020185

**Published:** 2022-01-11

**Authors:** Jing Du, Yingxue Lin, Yuan Gao, Yanyan Tian, Jixiang Zhang, Guozhen Fang

**Affiliations:** State Key Laboratory of Food Nutrition and Safety, Tianjin University of Science and Technology, Tianjin 300457, China; 13662198519@163.com (J.D.); lyx18322597673@163.com (Y.L.); 18222593097@163.com (Y.G.); tian1419556741@126.com (Y.T.); Cheng1213533550@163.com (J.Z.)

**Keywords:** rice, storage, nutrients changes, mildew warning

## Abstract

Processed unhusked rice is prone to mildew during storage. In this study, the storage conditions were simulated at temperatures of 20, 30, and 35 °C and a relative humidity of 40%, 60% and 80%, respectively. The water, fatty acid, and total starch content and the peak viscosity, mold colony number, protein secondary structure, and spatial structure of rice were monitored in order to propose the critical point of mildew during storage. In the process of rice from lively to moldy, the water content, fatty acid contents and the peak viscosity were increased. The total starch content decreased and then showed a slow increasing trend, while the microstructure of the powder particles changed from smooth and complete to loosen and hollow. With the increase in storage time, the vibration of the amide Ⅰ band of the rice samples decreased slightly, indicating that the total contents of β-fold, β-turn, α-helix, and random curl of the rice protein also changed. PCA (Principal Component Analysis) analysis showed that rice mildew index was closely related to temperature and humidity during storage. In our investigation, the best and most suitable temperature and relative humidity for rice storge is 20 °C and 40%, respectively. These results suggested that temperature and environmental humidity are vital factors affecting the physicochemical properties and nutrient changes, which provides a theoretical basis for the early warning of rice mildew during storage.

## 1. Introduction

Rice is refined through washing, peeling, rice milling, and finishing processes. It is an indispensable source of starch in the Chinese diet. According to statistics, more than 800 million people nationwide use rice as a staple food, and more than 85% of rice is consumed as a ration [1]. China’s total annual rice consumption is approximately 135 million tons [2]. At present, China’s grain reserves are mainly rice. In some rice-based distribution markets, ensuring the quality of rice during storage and delaying mildew are of great concern [3]. Compared to paddy, without the protection of rice husks, the storage conditions of rice must be more strictly controlled to avoid affecting the taste or the cooking color, or even mildew and moths [4].

Mildew is one of the biggest contributors to the loss of quality of paddy rice. It is estimated that the annual loss of rice caused by mildew in China is as high as tens of millions of tons. In the process of rice storage, the microorganisms carried by the air and the rice itself use the nutrients in rice to create conditions for its growth and reproduction, thus accelerate the aging and mildew of rice, resulting in mold [5]. In terms of the mildew of rice, some studies have reported that with the increase in storage time, the starch granules in rice change, resulting in a decrease in the viscosity of rice, a decrease in the iodine blue value, and an increase in water absorption [6]. The content of amylose soluble in hot water in rice decreases with increasing storage time, while the content of amylose insoluble in hot water gradually increases. Studies have pointed out that in rice stored for one year (20–25 °C), the amylose content increases by 4.94%, the insoluble amylose content increases by 4.26%, and the branched fixed flour content decreases by 6.05% [7].

During the storage of rice, fatty acids change via two ways: oxidation and hydrolysis. During oxidation, fatty acids are oxidized to produce peroxides, mainly aldehydes and ketones; during hydrolysis, fats are hydrolyzed by the lipolytic enzymes contained in the grain itself, and then glycerol and fatty acids are produced by molds [8]. During storage, fatty acids are difficult for microorganisms to utilize, resulting in an increase in the free fatty acid content. Therefore, the content of free fatty acids in grain has a good correlation with grain storage quality, which is one of the sensitive indexes to evaluate the deterioration degree of the grain quality [9,10].

The two fat changes of normal moisture grains can occur alternately or simultaneously [11]. Low moisture grains are mainly oxidized, and high moisture grains are mainly hydrolyzed [12]. In terms of proteins, during the storage period, grains will breathe weakly, and the proteins slowly hydrolyze under the action of hydrolytic enzymes, which will lead to a decrease in the protein content [11]. It is relatively easy to cause protein denaturation during grain aging. After protein denaturation, the spatial structure becomes loose, the peptide bond is extended, the non-polar group is exposed, and the hydrophilic group is embedded [13]. Under poor storage conditions, food is susceptible to microbial attack, and proteins are broken down into small molecules such as amino acids under the action of a series of enzymes secreted by the microorganisms, which reduces the protein nitrogen in the food, while non-proteins such as ammonia nitrogen and amine nitrogen increase the nitrogen level [7,14]. In the storage process, rice is susceptible to microbial contamination, increases in microbial substances secreted to decompose proteins into small molecular amino acids and other enzymes, decreases in protein nitrogen and ammonia nitrogen in rice, and increases in nitrogen amine nitrogen [12].

At present, the research on rice mainly focuses on its storage process from fresh to aged. However, aged rice enters the mildew stage, and the research on the changes in its nutritional composition is not yet mature. In this study, when rice was stored at temperatures of 20, 30, and 35 °C and a humidity of 40%, 60%, and 80%, the water, fatty acid, and total starch contents and the peak viscosity, mold colony number, protein secondary structure content, and spatial structure were monitored. Based on the changes in the mold colony quantity during rice storage, this study proposes the mildew critical point of rice storage, which plays a positive role in the early warning of mildew during rice storage. As a grain susceptible to mildew, it is vital to explore the change trend and mechanism of nutritional components such as starch, proteins, and lipids in the process of aging, which play a positive role in promoting the early warning of mildew in the process of rice storage.

## 2. Materials and Methods

### 2.1. Materials and Reagents

Rice was purchased from Tangshan City (Hebei Province, China) and harvested in October 2018; the time from harvest to milled rice is over 40 days. Anhydrous ethanol, KOH-(Potassium hydroxide) ethanol standard titration solution, phenolphthalein reagent, agar medium, potassium bromide, etc., were purchased from China National Pharmaceutical Group Chemical Reagent Co., LTD (Tianjin, China). The grade of the other regents and chemicals was analytical and used with no additional purification.

### 2.2. Experimental Equipment

An electrothermal constant temperature blast drying box (DL-101-1BS type) was purchased from Tianjin Zhonghuan Experimental Electric Furnace Co., LTD (Tianjin, China), an artificial climate box (MJ-250-II type) was bought from Shanghai Huitai Instrument Manufacturing Co., LTD (Shanghai, China), a scanning electron microscope (SU1510 type, Hitachi) was purchased from Tokyo (Japan), and an infrared spectrometer (tensor 27) was obtained from Bruker (Germany).

### 2.3. Experimental Methods

#### 2.3.1. Rice Storage

Three bags with an initial moisture content of 14.2% and weight of 2 kg were put into three separate sealed tanks. The samples were incubated in a constant temperature incubator-simulated warehouse storage, with a humidity of 40%, 60%, and 80% and a temperature of 20 °C. The initial water contents of the fresh samples were 13.5% and 13.7% at 30 °C and 35 °C, respectively [15].

#### 2.3.2. Sample Preparation

The sample preparation method was adopted from a previous study [16]. The sampling interval of the rice samples stored at 20 °C was set to seven days, 30 °C to five days, and 35 °C to three days. Then, 50 g of the stored rice samples were weighed, ground, and sifted through a 40-mesh sieve, and placed in a refrigerator at 4 °C before use.

#### 2.3.3. Measurement of the Moisture Content

The moisture content was determined by drying the rice to a constant weight at 105 °C (oven, Tianjin Zhonghuan Experimental Electric Furnace Co., LTD, Tianjin, China), (American Association of Cereal Chemists, 2000) [15].

#### 2.3.4. Measurement of Fatty Acid Content

The fatty acid value was determined using the official standard method (American Association of Cereal Chemists, 1976) [15]. The rice was pulverized and put through 40-mesh sieves for use. Then, 10 g of the pretreated rice sample powder was put into a conical flask, and 50 mL of anhydrous ethanol was added to shake it evenly, after which it stood for 1–2 min, and the filtrate was collected after filtration. The collected filtrate (25 mL) and 50 mL of distilled water were moved to another conical flask, and 3–4 drops of the phenolphthalein indicator were dropped in. The potassium hydroxide standard titration droplet was set to reddish, and the indicator liquid volume (mL) of the potassium hydroxide consumed was recorded. Anhydrous ethanol was used to replace the filtrate in the blank experiment. The fatty acid values (S) were expressed in milligrams of potassium hydroxide required to neutralize free amino acids in a 100 g dry matter sample, in milligrams per 100 g (mg/100 g).

#### 2.3.5. Changes in the Total Starch Content

The starch was isolated from rice using an alkaline steeping method with a minor modification [17]. Ten grams of pretreated rice sample powder was wrapped with filter paper and put into a Soxhlet extraction device. Petroleum ether was added and extracted at 45 °C for 10–12 h. After drying, defatted rice sample powder was obtained. Then, 50 mL of 0.1% NaOH solution was added to the sample, and the supernatant was removed after shaking for 30 min. The precipitation was washed and dried twice, and a defatted and deproteinized rice powder sample was obtained.

The defatted deproteinized rice powder sample (0.1000 g) and 10 mL of 1 mol/L KOH solution was added to a beaker; the beaker was placed in a boiling water bath and stirred thoroughly until the sample had completely dissolved. Then, 1 mL of the sample solution of constant volume was added into 30 mL of distilled water. The pH was adjusted to 3.5 with 0.1 mol/L of HCl solution, then 0.5 mL of iodine reagent was added to constant volume to 50 mL, after which it stood for 0.5 h for determination.

#### 2.3.6. Changes in Peak Viscosity

The peak viscosity of the milled rice was determined using the official standard method ISO 7973:1992 [18]. The rice was ground through a 60-mesh screen before use. The rice sample was weighed at approximately 3 g, and 25 mL of water was added in the sample barrel; then, the agitator was added to the sample barrel for 10 times to make the samples evenly dispersed. The mixer was nicely centered on the connector. The peak viscosity was recorded according to the viscosity curve displayed on the computer screen.

#### 2.3.7. Total Count of Mold Colonies

The total number of mold colonies during cultivation was determined using the national standard [19]. First, 25 g of the sample was weighed and fully shaken with 225 mL of sterile diluent was fully shaken to prepare a 1:10 sample homogenate. Then, 1 mL of the sample was absorbed into a test tube with 9 mL of sterile diluent and mix well. One mL of liquid was siphoned from the test tube into another test tube containing 9 mL sterile diluent, and so on, to prepare a 10-fold increasing series of diluted sample homogenates. Two or three sample homogenates with appropriate dilution were selected to absorb 1 mL into two sterile Petri dishes, and two 1 mL sterile diluents were absorbed into Petri dishes as a blank control. Then, 20–25 mL of potato dextrose agar was added to the Petri dish. After the agar had solidified, it was incubated in a 28 °C incubator (Shanghai, China) and counted.

#### 2.3.8. Scanning Electron Microscope Irradiation

The dried rice powder sample was adhered to a cylindrical base with a conductive adhesive, placed in a Hitachi E-1010 ion sputtering device, (Tokyo, Japan) and the coating time was set to 90 s. The acceleration voltage of the scanning electron microscope (Tokyo, Japan) was set to 15 kV and the beam current was 58 μA. The electron microscopy pictures were observed and photographed at different magnifications. A sample of micromorphology was recorded with a field-emission variable-pressure scanning electron microscope (SEM) (1530 VP, LEO, Oberkochen, Germany) [20].

#### 2.3.9. Infrared Spectrum Measurement

The potassium bromide powder was dried in an oven (Tianjin, China) at 110 °C for 3–4 h. Then, 150 mg of potassium bromide and 1 mg of dry rice powder were ground together in an agate mortar, which were loaded them into a mold and placed the mold on a tablet press (Tianjin, China), pressurized to 20 MPa. The pressure was released after holding for 1–2 min, and then the mold was taken out of the pressure. The thin slice was placed in the sample holder and infrared measurement was performed [21]. The results were preprocessed by OMNIC software (Thermo, Waltham, MA, USA) and then plotted by Origin 8.0 (OriginPro, Version 2021. OriginLab Corporation, Northampton, MA, USA), and fitted by Peak FIT software (V4.12, Systat Software, California, America).

### 2.4. Statistical Analysis

Changes in various indexes during rice storage were analyzed by a one-way analysis of variance. PCA was conducted to visually reflect the changes in the mildew index in rice under different storage conditions (different storage temperatures and humidity). Statistical analysis was performed using SPSS statistical software version 21 (SPSS, Inc., Chicago, IL, USA) for the analysis of variance.

## 3. Results

### 3.1. Changes in the Moisture Content

Figure 1 shows rice with an initial moisture content of 14.2% under the 20 °C storage condition. During the first sample measurement, it was found that the water content of the rice stored in an environment with a relative humidity of 40% was almost flat compared to fresh rice. Under the storage conditions of 60% and 80% humidity, the water content of the rice showed an increasing trend. At different storage temperatures, the moisture content increases gradually with the increase of ambient humidity. Presumably, this is because the lower the relative humidity, the slower the water balance between rice and the outside world, but the higher the relative humidity, the faster the water balance is completed, and the more the moisture content increases [22]. The initial moisture content of rice stored at 30 °C and 35 °C was different to that stored at 20 °C, but there was almost the same tendency in term of the water balance, and they all underwent the process of dynamic equilibrium. This phenomenon is consistent with the results of water content in Tao wang et al. [4].

### 3.2. Changes in the Fatty Acid Content

As an important standard to measure rice quality, the fatty acid content can be used as a sensitive index in the early warning stage of rice mildew, and it is also the main indicator for judging whether rice is aging [23,24,25]. The content of free fatty acids in the fresh rice samples was very small. The initial fatty acid content of rice was the highest when stored at 20 °C, which was 3.27 mg/100 g KOH. Moreover, the initial fatty acid contents of rice at 30 °C and 35 ℃ were slightly lower. The higher the temperature and the ambient humidity of rice storage, the faster the fatty acid value rises. When the ambient temperature is constant, the moisture content of the rice after equilibrium determines the increase in the fatty acid value [21]. As shown in Figure 2, at a storage temperature of 20 °C and an environmental humidity of 40% and 60%, the initial aging of rice lagged behind on the 35th and 40th days, respectively. In a high-humidity environment (80%), the fatty acid value slowly increased, and on the 35th day, the fatty acid value began to increase rapidly. When the temperature was 30 °C, the temperature at this time was the most suitable growth temperature range for fungi. At this temperature, the fatty acid content in three environments with different humidity rose rapidly. The same trend is observed at storage temperatures of 35 °C. A horizontal comparison showed that the higher the storage temperature, the faster the fatty acid content rose, and the earlier the initial stage of aging of the sample rice. These results were in agreement with previously published data on Food Chemistry [21,26]. It is inferred from the dynamic balance of water that this may be because the water content of rice had completed the first dynamic balance in the same period and had reached the maximum value, which was higher than the safe water content of rice [27].

At 20 °C and 80% of relative humidity, the fatty acid content of the rice samples showed a trend of first increasing and then decreasing. Due to the decomposition of fatty acids in rice, the water content of rice is reduced, and the value of fatty acids is also decreased accordingly [28]. When the ambient humidity was 60% and 80%, the turning point of the fatty acid value increase was at the 35th day, which was the same as the change of water content. However, when the ambient humidity was lower (40%), the growth rate remained relatively stable. This indicates that when stored at a low temperature of 20 °C, the change trend of the fatty acid value was relatively stable, while the content of fatty acid at a high temperature will increase significantly in a short time [29]. When stored at 30 °C and 35 °C, the content of fatty acids increased significantly with the increase of ambient humidity.

### 3.3. Changes in the Total Starch Content

In the process of reproduction, the rich starch material contained in rice is a good source of microbial energy, and becomes one of the essential conditions for its reproduction. As shown in Figure 3a, when the storage temperature is 20 °C, the relative humidity of the environment has little influence on microbial activities. As reflected in the figure, there was no significant difference in the reduction in starch content under three environmental humidity, but the microbial reproduction also caused a reduction in the starch content. However, at 30 °C, as shown in Figure 3b, the activity of microorganisms was more intense than that at 20 °C. Therefore, the higher the ambient relative humidity is at this temperature, the more the starch content decreases. The intense activity of microorganisms under this storage condition may bring adverse consequences in the process of rice storage. Figure 3c shows the change curve of the rice starch content when the temperature is 35 °C during rice storage. At this time, a large number of microorganisms take advantage of the nutrients in the rice, resulting in a reduction in the starch content. In addition, when the storage temperature is 30 and 35 °C, with the increase of environmental humidity, the utilization rate of nutrients in rice by microorganisms is significantly improved, and the starch content decreases. In the later stage of rice storage, a number of the microorganisms reached the saturation state, and the decrease rate of the starch content slowed down.

### 3.4. Changes in Peak Viscosity

Peak viscosity is an important index to measure the rice cooking quality and food quality, and its changes in storage can provide a certain theoretical basis for determining the degree of deterioration of rice quality during the storage process, with great significance for monitoring the rice quality during storage. As shown in Figure 4a, when rice was stored at 20 °C, the decrease in the microbial activity and enzyme activities of rice itself caused the peak viscosity of samples to change little, with only a slight upward trend. When stored at 30 °C, as shown in Figure 4b, the microorganisms resumed their activities and enzyme activity increased, increasing the peak viscosity with storage time. In the later storage period, when the environmental humidity is 80%, the peak viscosity decreases, which is equivalent to the environmental humidity of 40%, which may be affected by enzyme activity. When rice is stored at 35 °C, the peak viscosity increases more obviously with the change of humidity.

### 3.5. Changes in the Number of Mold Colonies

In the counting rule specified in the national standard, when the number of colonies is less than 10 CFU (Colony Forming Unit), it can be defaulted to 0 CFU. Under the initial conditions, the number of mold colonies in the fresh rice samples was close to 0 CFU [23]. Even if the mold was grown at a suitable temperature, the growth rate still lingered when the moisture content or ambient humidity had not reached the mold’s suitable growth humidity. It can be observed that in Figure 5, after the moisture content was balanced, the number of mold colonies under three environmental humidity increased dramatically. The higher the storage temperature, the more active the water dynamics, and the faster the growth and reproduction of mold.

When the storage temperature was 20 °C, the number of mold colonies of the stored rice samples in 80% relative humidity is significantly changed by the 40th day, which is the most suitable as a critical point for the early warning of mildew. For rice stored in two environments with a low relative humidity at 20 °C, the 50th day could be used as the critical point for the early warning of rice mildew. When the storage temperature rose to 30 °C, mold growth and reproduction were active at this time, and an obvious change in the number of mold colonies could be seen on the ninth day in a high-humidity environment, but the increase was small at this time. By the 15th–18th day, the temperature increase greatly advanced and caused the critical point of mildew to be greatly advanced during the storage of rice. Similarly, the critical point of mildew was also advanced to the 24th day in a lower relative humidity. When the storage temperature was 35 °C, the colony number of rice samples stored at 80% relative humidity changed significantly after 12 days of storage, and the mold growth activity was more active, while the 60% relative humidity showed no obvious change in colony number.

### 3.6. Analysis of the SEM

The main nutrients in rice are starch, protein, and fat in order from high to low. The content of starch in rice is up to approximately 75%. Taking rice stored at 20 °C and 80% relative humidity as an example, Figure 6a–I represents the micro-morphological structure of rice at different periods. It can be seen that with the increase in storage time and microbial activity, the starch granules gradually peeled off to form a single cube-shaped small granule. The fresh rice powder showed different sizes under the scanning electron microscope, and their surface was smooth and flat. The rice particles at this time had a tight combination of starch, proteins, and lipids. Encapsulations of starch, proteins, and the proteins are embedded as small molecular substances between the starch granules [10].

### 3.7. Analysis of Infrared Measurement

In addition to starch and lipids, the nutrients in rice also contain a small amount of protein. Due to the small proportion of protein, the content change during storage is not obvious, so only a rough study was made on whether it has changed. Proteins have several characteristic absorption bands in the infrared region, among which amide Ⅰ bands are valuable for studying the secondary structure of proteins. The characteristic peak of amide Ⅰ band is mainly stretching vibration of v C=O. The spectrum peak of amide Ⅰ band is confirmed to be relatively mature at present, in which 1650–1658 cm^−1^ is α helix, 1640–1610 cm^−1^ is β folding, 1700–1660 cm^−1^ is β corner, and 1650–1640 cm^−1^ is random crimping [24]. Therefore, in the infrared spectrum, the change in the vibration shrinkage rate of the amide I band of 1600–1700 cm^−1^ represents the secondary structure of protein [21,25,26]. Deconvolution and second-order derivation of amide I bands can be used to obtain changes in the spatial structure of rice proteins, such as β-fold, β-turn, α-helix, and random coiling. Figure 7 and Table 1, Table 2 and Table 3 shows the infrared spectra of rice samples with a relative humidity of 80% at 20 °C, 30 °C, and 35 °C. The box in the figure is the amide Ⅰ band. It can be seen that with an increase in storage time, the vibration of amide I band of sample slightly weakened, which shows that the β-fold, β-turn, α-helix, and total content of random coils is changed accordingly.

## 4. Discussion

### 4.1. Changes in the Moisture Content

Regardless of the temperature and relative humidity settings during storage, the moisture content of the rice stored under three relative humidity conditions showed a trend of increasing fluctuations during storage. During this period, the moisture content was greatly affected by the environmental humidity and the storage humidity. After a period of mutual balance with the culture environment, the growth and metabolism of the mold also produced metabolism water, trapped in the rice, causing the moisture content of the rice to increase, and making the moisture content of the rice reach equilibrium again [5,27]. Taking a sample with a storage temperature of 20 °C as an example, in an environment with a relative humidity of 40% the moisture content of the rice balanced to approximately 14.3%, with a relative humidity of 60% the moisture content is balanced to about 14.5%, and with a relative humidity of 80% the moisture content is balanced to about 14.6%. During storage, an increasing trend in the fluctuations was observed. However, after the first round of equilibrium has been completed, due to the use of microorganisms and the relative effects of the external environment, the moisture content showed a downward trend. The higher the environmental humidity, the faster the first round of the equilibrium is completed, and the faster the subsequent decrease in moisture content.

### 4.2. Changes in the Fatty Acid Content

Studies have shown that when the fatty acid content of rice reaches 8 mg KOH/100 g, the shelf life of rice is over; in other words, an unpleasant aroma and texture form in the rice [28]. The higher the temperature and the ambient humidity of rice storage, the faster the fatty acid value rises. When the ambient temperature is constant, the moisture content of the rice after equilibrium, determines the increase in the fatty acid value [29]. Under unsuitable conditions, such as high-temperature and high-humidity storage conditions (35 °C and 80% relative humidity, respectively), lipolytic bacteria can rapidly grow in rice. Such bacteria in rice include *Aspergillus flavus*, *Aspergillus Niger*, *Aspergillus fumigatus*, *Penicillium griseorum*, *Penicilliu chrysgenum*, *Mucor lipolytica*, *Geotrichum candidum*, *Candida lipolytica*, *Bacillus*, *Pseudomonas*, *Micrococcus*, etc. [5,30]. These microorganisms can decompose fat to produce the primary product of higher fatty acids, which cannot be used by microorganisms and, thus, accumulates in rice, causing the fatty acid content to gradually increase and the rice to age [31].

When stored at a low temperature of 20 °C, the change trend of the fatty acid value was relatively stable, while the content of fatty acid at a high temperature will increase significantly in a short time [32]. When the storage temperature was raised to 30 °C, the content of fatty acids increased significantly under high-humidity conditions. In Figure 2b, the fatty acid value of the rice increased more obviously from the 15th day. It is recommended that grain piles and warehouses should be ventilated and cooled down on the 15th day to delay the increase in the fatty acid worth. A temperature of 35 °C, the temperature at this time is not suitable for long-term storage of rice [33]. If manual intervention is not carried out, serious economic losses will occur. When the rice enters the moldy stage after aging, under the action of glycerol kinase, glycerol is absorbed by microorganisms to produce α-glycerol phosphate, which is then dehydrogenated to form a mainstream metabolic pathway (EMP-TCA) for complete oxidation of dihydroxyacetone phosphate. Conversely, dihydroxyacetone phosphate can also react reversely along EMP to synthesize carbohydrates in microbial microorganisms [11,28,34].

### 4.3. Changes in the Total Starch Content

Changes in the starch content during storage have been reported [14]. Our results also shown that during rice storage, a change in starch content can be used as the critical point to judge mildew. When the temperature was 20 °C, if the grain pile was in a high humidity environment, the microbial growth was slow, and the starch content was basically in a stable state. When rice is stored at 30 °C and in a high-humidity environment, although the temperature is not suitable temperature for microbial growth, samples should be tested on the 18th day and certain measures should be taken. Rice should be stored at 35 °C, and the environment and samples should be kept dry. In the experiment, the first decline in the starch content appeared at the earliest on the eighth day, and the number of microorganisms in the grain pile reached the first saturation point, which may cause grain pile deterioration and harm the edible quality of rice.

### 4.4. Changes in the Peak Viscosity

Similar to other indexes, the peak viscosity of rice tends to rise first and then flatten. The reason may be that during the storage process, the α-amylase activity in the rice decreased continuously—the higher the temperature, the higher the relative humidity of the environment, the wetter the rice, and the more the α-amylase activity decreased [35]. Due to the control of temperature and humidity in the storage environment, the amylose content of the rice increased, while the amylopectin content decreased. The crystal structure of starch grains was affected by the change in the ratio between the two starches, leading to a corresponding change in the internal structure of rice grains. This is consistent with the research results of Chang-Jie Yan et al. [36]. The ability of water transformation in grains improved, the speed of water entering into the grains was accelerated, and the gelatinization temperature of the rice was increased [37].

Peak viscosity has a great influence on starch structure, starch content, and rice α amylase content. In the process of storage, the temperature and humidity of the change does not directly impact on the peak viscosity, but instead affect the structure. Therefore, it is not recommended to take the peak viscosity of rice as a direct indicator of the early warning of rice mildew, but as a reference combined with other more sensitive early warning indicators to predict rice mildew.

### 4.5. Changes in the Number of Mold Colonies

It can be observed in Figure 5 that the relative humidity of 80% was the closest to the optimal growth humidity of mold [18], so the rice microorganisms stored under this condition were the most active and had the greatest colonies. Similarly, the samples in the high-humidity environment showed a trend of increasing colony numbers first and then decreasing at the advanced stage of cultivation. This happened because there is a competitive relationship in the growth process of microorganisms. In the early stage, when the nutrients are rich, the temperature and humidity reach the optimal conditions and the microorganisms begin to multiply rapidly. After a period of time, the quantity reaches saturation [38]. At this time, the number of nutrients available for molds is greatly reduced; that is, there are few nutrients but many microorganisms, which causes some molds to continue to reproduce or even not to survive, so the number of mold colonies appears to decrease.

The mildew of rice is caused by the growth and reproduction of mold, while the increase of mold quantity is closely related to water content. We reasonably believe that the order of magnitude of the increase in the number of mold colonies can be used as an important indicator to judge whether there is a tendency of mildew in rice. Jiayi Shi also found that the decrease in moisture may have been the reason for mold growth inhibition [23].

### 4.6. Analysis of the SEM

During storage, starch provides rich carbohydrates for microbial growth. When the rice enters the aging stage, the lipid shell is damaged by microbial activity, exposing the starch particles therein [39]. Rice is stored under the optimal growth conditions for microorganisms. Microorganisms continue to use starch, lipids, proteins, and other nutrients, causing the structure of rice powder particles to change, and the lipid shell to fall off. When molds multiply and the microscopic colonies appear on the surface of the rice, voids can clearly be seen in the microstructure of the rice, and the structure becomes looser [40]. From Figure 6, it can be concluded that during the process of rice from fresh, aged to mildew, the microstructure of its powder particles changed from smooth to loosen voids.

### 4.7. Analysis of Infrared Measurement

In the storage process, with the increase in time, the relative content of α-helix, β-turn, β-sheet, and random coiling fluctuated without specific laws. The stability of the α-helix structure is closely related to non-covalent bonds, such as hydrogen bonds, which can be influenced by temperature, pH, and other factors. Such changes may be caused by microbial activity [27]. In the later period of storage, microorganisms multiply rapidly, and they consuming a large amount of nutrients in the rice, so that they can be converted into the energy which they need for their growth [21]. Therefore, the moisture content, fatty acid, starch, and protein contents are affected to varying degrees.

### 4.8. Study on the Early Warning of the Mildew Temperature and Humidity of Rice

Mold growth and reproduction need a specific temperature and humidity; generally speaking, the optimal growth of mold temperature is 28–33 °C and the optimal growth humidity is about 70% [23]. When there is a mutation point in the total number of mold colonies in stored rice, the storage environment temperature and humidity, which are obviously correlated with it, will change from the previously stable state to an abnormal fluctuation state. When the ambient temperature rises to 30 °C, the abnormal turning point of the ambient temperature and humidity curve is earlier than that at 20 °C. The original 30–40 days period is shortened to 15–20 days, and the time of the abnormal point of the ambient temperature and humidity fluctuation is corresponding to that shown in Figure 5b.

Although the temperature of the simulated chamber was set at 35 °C, the actual monitored temperature was 33 °C due to the limitation of environmental facilities. At this time, the storage temperature is the most suitable temperature for the growth of mold colonies. When the optimal environmental humidity is satisfied, mold can grow and reproduce rapidly in the rice pile. It can be seen from Figure 5c that the optimum temperature for rice piles changed under a high humidity environment and the storage temperature, humidity, and temperature continued to rise, resulting in the “heartburn” phenomenon in rice. The rice changed from a fresh dispersed particle to block, and its color and smell also changed. Without artificial control, gray-green mold colonies could be seen in the rice pile with the extension of storge time, resulting in a serious waste of rice.

The total number of mold colonies is the most intuitive and accurate critical point for the early warning of mildew during rice storage. However, this early warning point has the shortcoming of hysteresis [41]. Therefore, the temperature and humidity changes of the storage environment can also be taken into account, so that the critical point of early warning of mildew in the storage process of rice is sensitive, rapid, and accurate. When the turning point of the temperature and humidity changes in the storage environment is used as the early warning point for mildew of rice, as long as the temperature and humidity are monitored in real time during the storage process, and the data recording interval should not be too long to avoid hysteresis and missing the best adjustment opportunity. Once the inflection point of the temperature and humidity curve is found to be abnormally high, the mildew warning can be carried out.

### 4.9. PCA Analysis

Principal component analysis was performed on the important factors affecting the mildew of rice samples. The first two principal components (PCs) accounted for 73% of the variation (Figure 8a,b). PC 1 and PC 2 accounted for 42.2% and 30.8% of the total variance, respectively. Rice samples were separated into nine clusters according to the storage temperature and humidity. As the storage proceeded, samples were spread on the positive side of PC 1 and the positive side of PC 2. The mildew index in rice samples was correlated with temperature and humidity during storage. The storage environment at 20 °C is closely related to fatty acid content, water content, and total starch content, the storage environment at 30 °C is highly related to the number of mold colonies, and the storage environment at 35 °C is significantly related to peak viscosity.

## 5. Conclusions

During the storage of rice samples, when the storage temperature and humidity were in the optimal growth range of mold, the moisture content, fatty acid content, total starch content, peak viscosity, total colony number, starch particle shape, and protein spatial structure content of the rice samples all changed to different degrees. Storage temperature and humidity are important factors affecting mildew. As the growth of microorganisms requires the utilization of the nutrients in rice, the microstructure of rice flour grains changes from smooth and intact to loosen and hollow with the increases in storage time. With the increase in sampling times, the vibration of the amide Ⅰ band in the rice decreased slightly. The total contents of β-folding, β-turning, α-helix, and random helix in rice protein also changed correspondingly. PCA analysis showed that rice mildew index was closely related to temperature and humidity during storage. The change trend of the fatty acid value and the rapid increase in the colony number provided a warning for mildew in rice storage. This study confirmed the mildew rule of rice at a specific storage scale, laying a foundation for preventing and controlling mildew in the process of rice storage.

## Figures and Tables

**Figure 1 foods-11-00185-f001:**
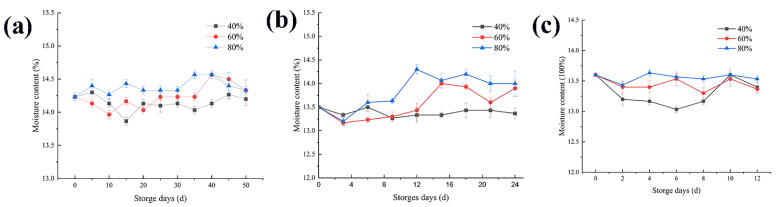
Changes in the rice moisture content at different storage temperatures: (**a**) 20 °C, (**b**) 30 °C, and (**c**) 35 °C.

**Figure 2 foods-11-00185-f002:**
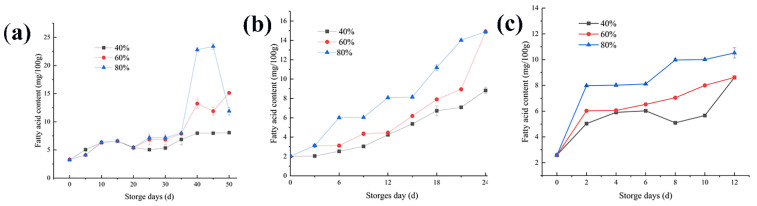
Changes in the rice fatty acid content at different storage temperatures: (**a**) 20 °C, (**b**) 30 °C, and (**c**) 35 °C.

**Figure 3 foods-11-00185-f003:**
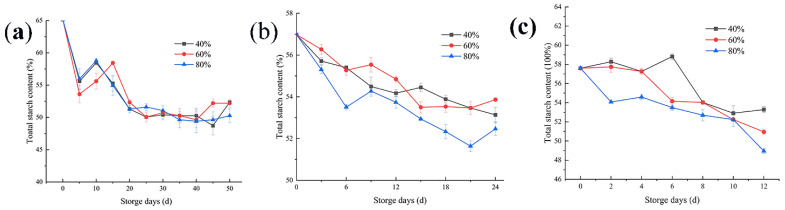
Changes in the total starch content of rice under three different relative humidity conditions at storage temperatures of 20 (**a**), 30 (**b**), and 35 °C (**c**).

**Figure 4 foods-11-00185-f004:**
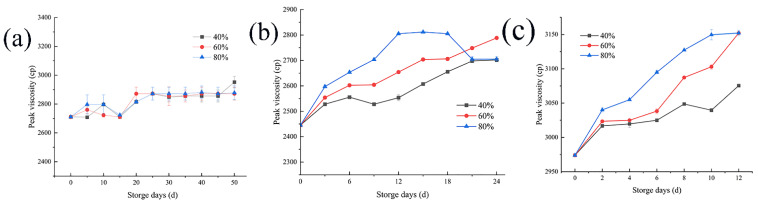
Peak viscosity of rice at storage temperatures of 20 (**a**), 30 (**b**), and 35 (**c**) °C under three different environmental relative humidity.

**Figure 5 foods-11-00185-f005:**
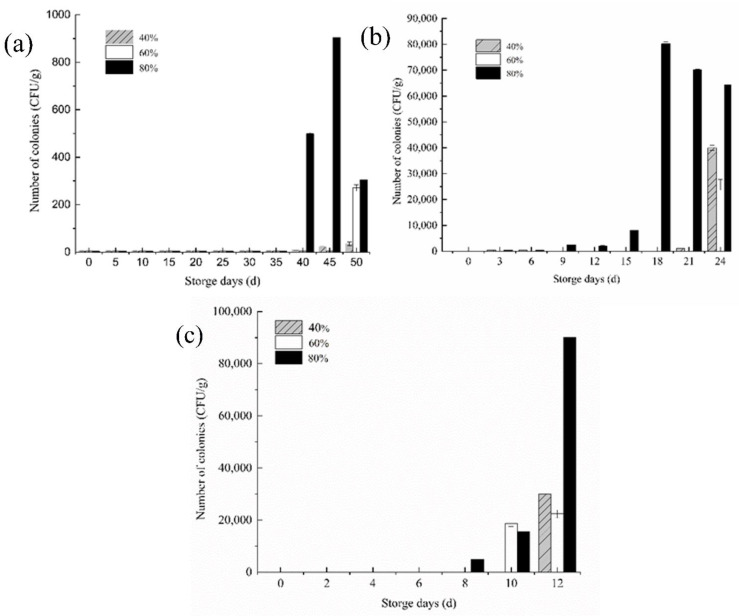
Changes in the number of rice mold colonies at different storage temperatures (**a**) 20 °C, (**b**) 30 °C, and (**c**) 35 °C.

**Figure 6 foods-11-00185-f006:**
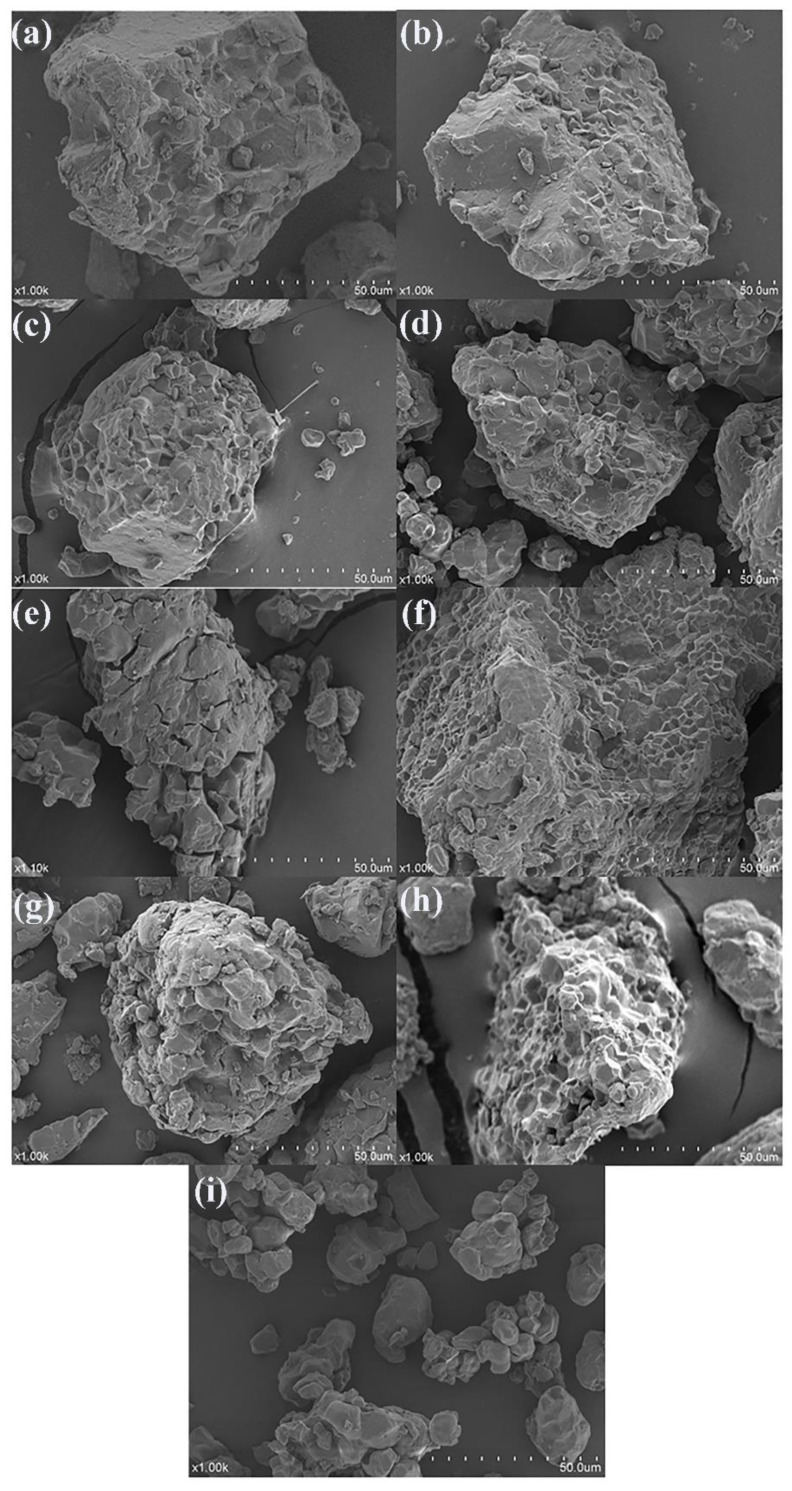
Electron microscopic morphology of the rice samples stored at 20 °C and humidity at 80%: (**a**) 0th day of storage; (**b**) 15th day of storage; (**c**) 20th day of storage; (**d**) 25th day of storage; (**e**) 30th day of storage; (**f**) 35th day of storage; (**g**) 40th day of storage; (**h**) 45th day of storage; (**i**) 50th day of storage.

**Figure 7 foods-11-00185-f007:**
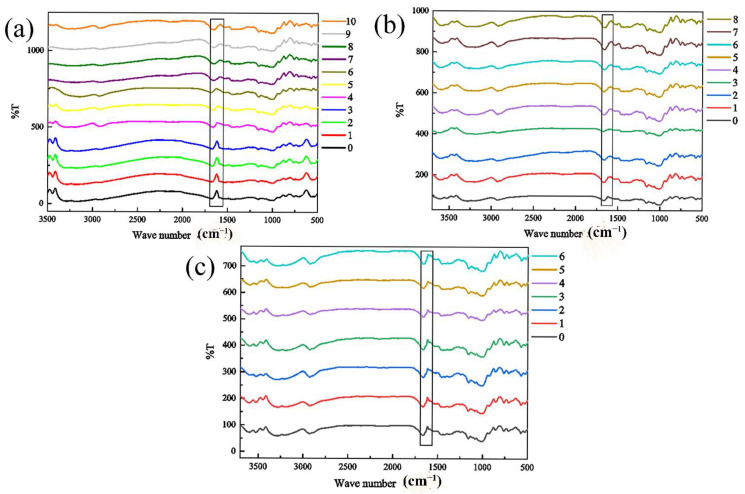
The change in the infrared spectrum of rice under ambient humidity of 80% different storage temperatures (**a**) 20 °C, (**b**) 30 °C, and (**c**) 35 °C.

**Figure 8 foods-11-00185-f008:**
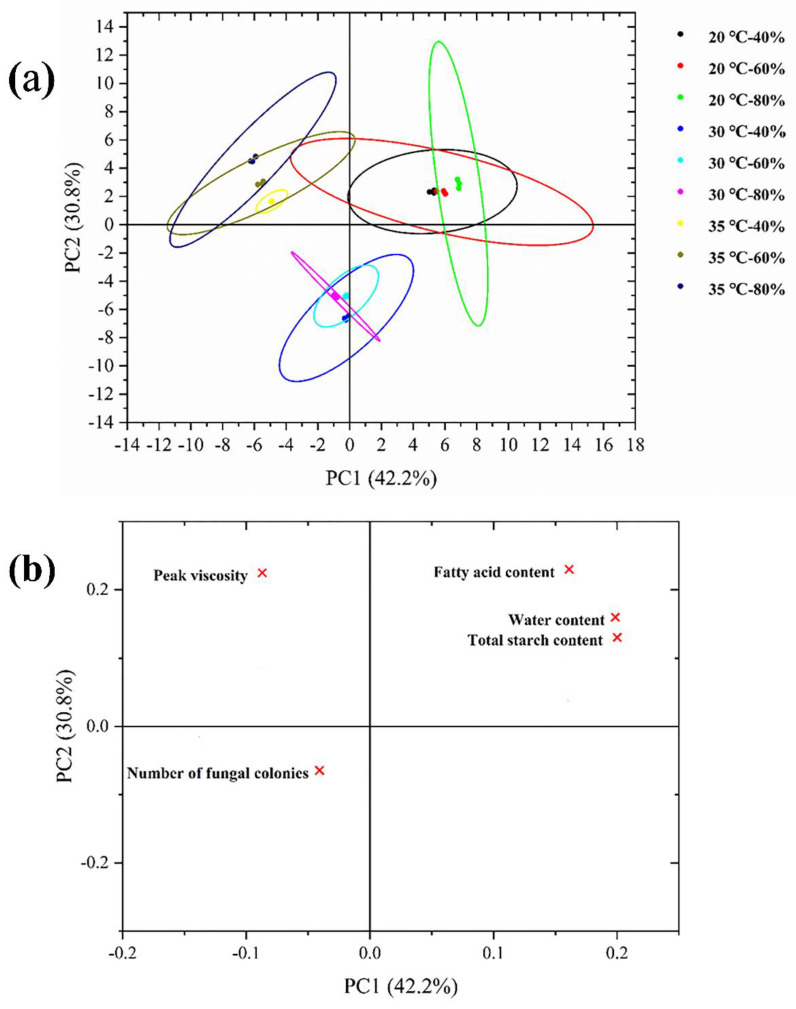
(**a**) PCA score plot and (**b**) PCA loading plot based on the critical point of mildew in stored rice. Note: × represents an important index in the process of rice mildew.

**Table 1 foods-11-00185-t001:** Changes in the spatial protein content of rice protein when the relative humidity is 80% and the storage temperature is 20 °C.

Storge Days	α-Helix	β-Sheet	β-Turn	Random Coli
0 days	54.61%	12.23%	14.53%	18.63%
1 days	49.81%	10.64%	12.36%	27.21%
2 days	54.12%	13.58%	14.67%	17.63%
3 days	47.96%	12.15%	14.62%	25.27%
4 days	55.32%	13.17%	15.63%	15.88%
5 days	53.63%	12.89%	13.98%	19.50%
6 days	48.56%	15.26%	13.54%	22.64%
7 days	50.23%	14.89%	13.95%	20.93%
8 days	53.21%	11.78%	12.10%	22.91%
9 days	55.69%	14.78%	14.62%	14.91%
10 days	54.30%	13.58%	15.21%	16.91%

**Table 2 foods-11-00185-t002:** Changes in the spatial protein content of rice protein when the relative humidity is 80% and the storage temperature is 30 °C.

Storge Days	α-Helix	β-Sheet	β-Turn	Random Coli
0 days	53.16%	18.32%	11.54%	16.98%
1 days	52.64%	15.26%	10.98%	21.12%
2 days	54.65%	17.62%	13.17%	14.56%
3 days	49.85%	20.16%	14.52%	15.47%
4 days	52.17%	16.13%	12.85%	18.85%
5 days	50.64%	13.69%	11.27%	24.40%
6 days	51.67%	11.25%	13.42%	23.66%
7 days	51.29%	12.56%	15.86%	20.29%
8 days	54.39%	13.18%	15.26%	17.17%

**Table 3 foods-11-00185-t003:** Changes in the spatial protein content of rice protein when the relative humidity is 80% and the storage temperature is 35 °C.

Storge Days	α-Helix	β-Sheet	β-Turn	Random Coli
1 days	49.25%	13.14%	16.58%	21.03%
2 days	52.17%	11.89%	14.00%	21.94%
3 days	52.86%	13.65%	15.77%	17.72%
4 days	50.29%	11.64%	14.89%	23.09%
5 days	53.02%	12.15%	15.77%	19.06%

## Data Availability

Not applicable.

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
