# Peer review of "Nutritional Changes and Early Warning of Moldy Rice under Different Relative Humidity and Storage Temperature"

_foods, 2022, doi:10.3390/foods11020185_

Round 1
Reviewer 1 Report
I accept the corrected and amended version
Author Response
Thank you for your constructive comments in the process of revising the article. We really appreciate the reviewer’s hard work on looking through our paper. The reviewers’ comments are very helpful in our following experiments. We really thank the comments from this reviewer!
Reviewer 2 Report
The ms foods-1527636 entitled Nutritional changes and early warning of moldy rice under different relative humidity and storage temperature investigated the storage conditions of moldy rice at temperature of 20, 30, and 35 ℃ and relative humidity of 40%, 60% and 80%, respectively. The water, fatty acid, and total starch content and the peak viscosity, mold colony number, protein secondary structure and spatial structure of rice were monitored in order to propose the critical point of mildew during storage was proposed. The ms was improved, but here are some additional comments:
L21-23: In the conclusion at the end of the abstract> Please write directly the best and suitable temperature and relative humidity in your investigation, based on your findings. Do not make it general and superficial.
L57-62: who said this? Please cite relevant references!
Why the authors decided to study the effects of temperature 20, 30, and 35 ℃ and humidities of 40%, 60% and 80% on rice? Why not other levels? Why not 20, 30, and 40 ℃? Why 35℃? The increasing was 10 then 5!
The authors must cite the described methods in material and methods section! Add the references of those methods, I am sure that they are not your own.
Some results were not discussed in the discussion part. If possible, the author can divide the results from the discussion.
Good luck, Reviewer
Reviewer 3 Report
Dear Authors,
thank you for revising the manuscript at a linguistic and scientific level, and providing a new version of your manuscript.
I have one more suggestion. The discussion of your results upon comparison with other studies available in literature is still a bit poor. I kindly ask you a further effort, so as to improve further the quality of your paper. Try to comment on your data by discussing if they are in line with data from other investigations or if you got different results.
Author Response
Thank you for your constructive comments. We have revised the discussion in the revised manuscript. We really appreciate the reviewer’s hard work on looking through our paper. The reviewers’ comments are very helpful in our following experiments. We really thank the comments from the reviewer! We will keep revising the paper until it reaches the standard of Foods.
Thanks for your valuable suggestion. We apologize for these faults. We have tried comment on some data by discussing our results and comparing with other studies available in literature in revised paper.

This manuscript is a resubmission of an earlier submission. The following is a list of the peer review reports and author responses from that submission.
Round 1
Reviewer 1 Report
Review: foods-1477359
The manuscript tracks changes in rice composition as a function of storage temperature, humidity and microbial growth.
I think the idea is a good one because monitoring changes during storage is an important issue.
However, I do not feel that the implementation of this good idea is of sufficient quality. The repetition of measurements and the spread of repetitions is lacking.
The measurement results are practically not evaluated. A higher or slightly lower "evaluation" is not meaningful.
The presentation of testing methods is weak.
Neither the SEM nor the equipment used for the infrared measurements is described.
The evaluation of the infrared spectra is very poor and incomplete.
The manuscript has so many shortcomings that I do not consider it suitable for publication in a high quality journal such as Foods

Reviewer 2 Report
The ms foods-1477359 with the title of Nutritional changes and early warning of moldy rice under different relative humidity and storage temperature investigated rice storage conditions under 20 ℃, 30 ℃ and 35 ℃ and at 40%, 60% and 80% relative humidity. The ms is well organized, however here are some comments that make the ms stronger and suitable for publication in Foods. There are some typos errors, and the authors have to correct all those issue.
L11-15 very long sentence, please divide it into two sentence. Also, try to avoid repeating the rice many times per sentence. I noted that you repeat (rice) more than once per sentence.
L15-17 correct this sentence, something is wrong>> Why you repeated (from living to moldy) twice. Check what I suggested here>> In the process of rice from lively to moldy, water content, fatty acid content and peak viscosity were increased.
L16 or L18 you can not make the letter after the comma as capital. It is wrong>> water not Water >>> the not The
L25 barley??? What do you mean? Barley it is a plant! Or do you mean the full grain of rice is called barley?
L28-35 who said all this text? Of course it is your own words and it is not your own numbers and statistics, so please add relevant references to this text.
L46 leave space 6.05%[5].
L47-52 who said this, add relevant reference
L62 and 65 leave space ded[8]. gen[9,5].
L67-69 the aim of this study should be clear as it is written in the abstract. Please revise it.
Section 2.3.4. Authors have to describe the method of Measurement of fatty acid content in details.
Where is the discussion section
Although the results section is well written and organized with Figures, but my concern is: There is not statistical analysis for the presented data! Why? How I can say that this treatment is better than that treatment?
Conclusion should be shorter and the author should focus on the most important findings.
Reviewer 3 Report
The authors have made great efforts to investigate the changing trend and contents of rice starch, protein, and lipid during the process of mildew. The results are of interest to improve the quality of rice during storage. However, the current paper has not been well documented and is poorly organized. It needs to be critically revised. Some comments and suggestions may be useful to improve the paper as below:
- Introduction must be improved (see in comments in pdf file)
- Materials and Methods should be totally reworded (see in the file).
- Lack of Discussion part, the attained results have been conflicted with the methods (see in the file)
- All figures are unclear, should be made with high quality and solution
- Conclusion is lengthy, should be shortened

Reviewer 4 Report
The paper “Nutritional changes and early warning of moldy rice under different relative humidity and storage temperature” aims to report on the main results of a study investigating the effect of storage conditions on rice nutritional profile and mildew attitude.
The manuscript is not suitable for publication at the current state. However, the Authors can find below general and specific remarks for improvement.
As general remarks:
- The manuscript requires an extensive revision of the English language, otherwise the scientific value is deeply affected by the poor English language and style.
- The structure and the organization in paragraphs also require an optimization/improvement. Especially in the section Materials and Methods.
- In the Section “Results and Discussion”, an extensive revision is also necessary. Because of the language/style used, the reader is not well aware of the results observed/obtained in the study and the general considerations made by the Authors. You must therefore work extensively on this section. In addition, more studies currently available on literature should be used and mentioned in order to discuss the obtained results.
Please, find below some points which require the attention of the Authors.
Abstract
Lines 11-15: please, amend this sentence and make it shorter and clearer. You might split it into two sentences, for instance.
Introduction
Lines 25-26: make this sentence clearer.
Lines 28-29: please, provide a reference for this sentence “China's total annual rice consumption is about 135 million tons”.
Lines 34-35: please, provide a reference for these data.
Lines 38-39: please, check the sentence.
Line 67: check English, please.
Materials and Methods
Lines 72-76: this part requires an extensive revision. Please, amend it so that it is in good English language and does not sound as a list.
Lines 78-81: as per lines 72-76. Please, do not provide a mere list of equipment but organize all the information in proper sentences.
Lines 84-86: please, reformulate these sentences and use the passive tense. Please, do not report them as if you were giving instructions.
Lines 89-91: please, reformulate these sentences and use the passive tense. Please, do not report them as if you were giving instructions.
Lines 93-94: Please, specify the method number.
Lines 96-97: Please, specify the method number.
Lines 128-130: please, reformulate these sentences and use the passive tense. Please, do not report them as if you were giving instructions.
Results and discussion
Lines 171-173: please, use italics for microorganisms names.
Line 179: please, provide a reference.
Lines 182-193: Please, add references to the Figure you are discussing.
Lines 195-210: Please, amend this part so that it is clear to the reader when you are making a statement regarding what you observed, and when you are making general considerations. To accomplish this, add a reference to your figures and refer to literature when you are making general statements.
Lines 211-222: the same than above applies. Please, reformulate the whole section.
Lines 236-243: Please, amend this sentence. Split it into more than one single sentence.
Lines 246-260: I suggest removing the 1), 2) and 3). Try to work on the style and avoid it. Also revise the sentences to make the results clearer.
Lines 276-281: please, amend this sentence and make it clearer. Split it into more than one single sentence.
Lines 349-351: please, provide references.
References
Amend the style according to the Journal guidelines.